# An improved sea ice detection algorithm using MODIS: application as a new European sea ice extent indicator

Joan Antoni Parera-Portell[1,*], Raquel Ubach[1], and Charles Gignac[2]

[1]Department of Geography, Univesitat Autònoma de Barcelona, Barcelona, Spain
[2]TENOR laboratory, Institut National de la Recherche Scientifique - Centre Eau Terre Environnement, Quebec City, Canada
[*]Now at: Instituto Andaluz de Geofísica y Prevención de Desastres Sísmicos, Universidad de Granada, Granada, Spain

**Correspondence:** Joan Antoni Parera-Portell (jpareraportell@ugr.es)

**Abstract.** The continued loss of sea ice in the Northern Hemisphere due to global warming poses a threat on biota and human activities, evidencing the necessity of efficient sea ice monitoring tools. Aiming at the creation of an improved sea ice extent indicator covering the European regional seas, the new IceMap500 algorithm has been developed to classify sea ice and water at a resolution of 500 m at nadir. IceMap500 features a classification strategy built upon previous MODIS sea ice extent algorithms

and a new method to reclassify areas affected by resolution-breaking features inherited from the MODIS cloud mask. This approach results in an enlargement of mapped area, a reduction of potential error sources and a better delineation of the sea ice edge, while still systematically achieving accuracies above 90 %, as obtained by manual validation. Swath maps have been aggregated at a monthly scale to obtain sea ice extent with a method that is sensitive to spatio-temporal variations of the sea ice cover and that can be used as an additional error filter. The resulting dataset, covering the months of maximum and minimum

sea ice extent (i.e. March and September) during two decades (from 2000 to 2019), demonstrates the algorithm's applicability as a monitoring tool and as an indicator, illustrating the sea ice decline at a regional scale. The European sea regions located in the Arctic, NE Atlantic and Barents seas display clear negative trends both in March ($-27.98\pm6.01\times10^3$ km$^2$yr$^{-1}$) and September ($-16.47\pm5.66\times10^3$ km$^2$yr$^{-1}$). Such trends indicate that the sea ice cover is shrinking at a rate of $\sim$9 % and $\sim$13 % per decade, respectively, even though the sea ice extent loss is comparatively $\sim$70 % greater in March.

*Copyright statement.* TEXT

## 1 Introduction

The Arctic sea ice cover has been changing rapidly over the last decades, with its overall extent declining steadily since the first satellite observations in the late 1970s (e.g. Cavalieri and Parkinson, 2012; Massonnet et al., 2012; Meier et al., 2014; Parkinson, 2014; Serreze and Stroeve, 2015; Comiso et al., 2017), reaching its historical minimum on September 2012. The

same decreasing trends are also evidenced by other parameters such as sea ice thickness (Kwok, 2018; Liu et al., 2020), which has decreased as much as 65 % in the period extending from 1975 to 2012 (Lindsay and Schweiger, 2015). This massive loss of ice is unprecedented in the last few thousand years (Polyak et al., 2010), and is attributed both to climatic variability and to

external forcing caused by the anthropogenic release of greenhouse gases (e.g. Myhre et al., 2013; Stroeve and Notz, 2018). All projection models agree that Arctic sea ice will continue shrinking and thinning, eventually leading to ice-free summers in the upcoming decades (Massonnet et al., 2012; Stroeve et al., 2012; Collins et al., 2013; Notz and Stroeve, 2016; Stroeve and Notz, 2018) and even as soon as in the late 2030s (AMAP, 2017).

The dynamism of the sea ice and the effect it has on climate, biota and human activities makes the regular monitoring of its properties (e.g. extent, concentration, thickness) necessary. Sea ice data is nowadays continuously obtained from several satellite-borne instruments (e.g. Spreen and Kern, 2016), among which microwave sensors stand out for their ability to acquire data in disregard of the lighting and weather conditions. Passive microwave sensors typically provide data at resolutions above 15 km, hindering their use for local and regional sea ice studies. On the other hand, active microwave and visible-infrared sensors can acquire data at much higher spatial resolutions. For instance, ESA's satellites Sentinel-1 (synthetic aperture radar) and Sentinel-2 (visible-infrared) achieve resolutions of 5-100 m in the first case, and 10-60 m in the latter. However, such high-resolution sensors render data with sparse spatial and temporal coverage due to their small swath size and long revisit times, although this effect is minimized at the poles. Instead, MODIS visible and infrared imagery offers a balanced trade-off between temporal and spatial coverage. MODIS is an imaging sensor on board of NASA's sun-synchronous satellites Terra and Aqua, launched in 1999 and 2002, respectively. It acquires data in 36 spectral bands, ranging from the visible spectrum to the thermal infrared. Spatial resolution at nadir varies from 250 m (bands 1 and 2) to 500 m (bands 3-7) and 1 km (bands 8-36), and has a large swath width of 2330 km. The MODIS Terra and Aqua MOD29 and MYD29 datasets (Hall et al., 2015a,b) provide daily global sea ice extent coverage at 1 km, but frequently fail to map the sea ice edge at this level of detail. This is caused by the MODIS MOD35_L2 cloud mask product (Ackerman et al., 2010; MODIS Atmosphere Science Team, 2017), the accuracy of which depends on the correct identification of background sea ice at 25 km resolution (Riggs and Hall, 2015). Therefore, sea ice beyond this background is finally labelled as cloud instead of clear, eventually preventing the products which rely on this cloud mask from accurately mapping the sea ice cover.

In this context, a new 500 m resolution MODIS sea ice detection algorithm (IceMap500) was developed, aiming at the improvement of existing European sea ice extent indicators based on passive microwave observations (EEA, 2020) by providing additional and higher resolution data. IceMap500 is heavily influenced by the cloud masking and classification approaches of the previous IceMap250 algorithm (Gignac et al., 2017), which nonetheless is still vulnerable to the MOD35_L2 background effects. The new algorithm is optimized to minimize classification errors, and improves the quality of the maps by introducing a five-step workflow to prevent MOD35_L2 from breaking the 500 m resolution.

To test the usefulness of IceMap500 as a European sea ice extent indicator we analyse the sea ice trends in the European regional seas from 2000 to 2019 using MODIS Terra data. The analysis covers the northernmost European sea regions defined by the European Union's Marine Strategy Framework Directive (MSFD) where sea ice might occur (EEA, 2018), and is restricted to the months when the maximum and minimum sea ice extent is reached in the Northern Hemisphere, that is, March

and September, respectively.

## 2 Materials and methods

### 2.1 Study area

This work focuses on the European regional seas established by the MSFD (EEA, 2018). As sea ice only occurs in the northern-most oceanic sea regions or in enclosed, low-salinity water bodies such as the Baltic Sea, spatial coverage has been significantly reduced to avoid the processing of uninformative data. The final study area extends over the sea regions in Fig. 1, covering an area roughly $4 \times 10^6$ km$^2$. With the inclusion of a 400 km buffer to coherently join all the target regional seas in a single study region, the totality of the processed area sums up to approximately $8 \times 10^6$ km$^2$.

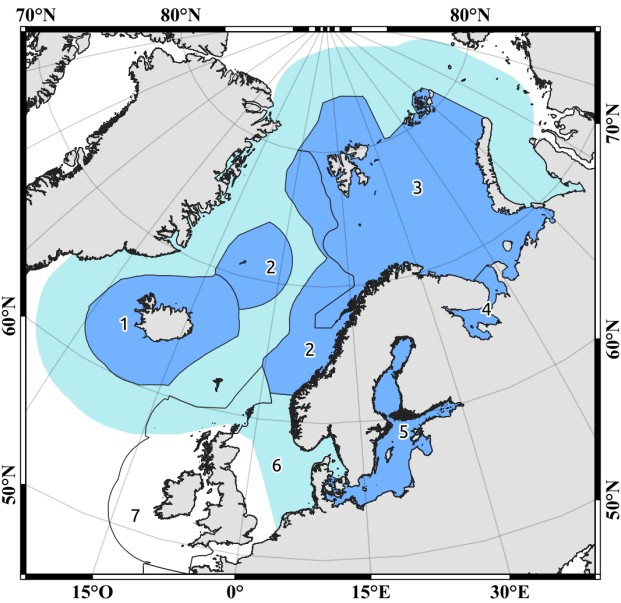

**Figure 1.** Northern European regional seas, as defined by the MSFD: 1) Iceland Sea, 2) Norwegian Sea, 3) Barents Sea, 4) White Sea, 5) Baltic Sea, 6) Greater North Sea, and 7) Celtic Seas. In medium blue are shown the target sea regions, whereas in light blue is represented the generated buffer, whose external limit corresponds to the total processed area. All maps in this work are shown in North Pole Lambert Azimuthal Equal Area.

Oceanic sea ice in the Northern Hemisphere has both a perennial and a seasonal fraction. Typically, maximum and minimum sea ice extent are reached in March and September (e.g. Stroeve et al., 2008), respectively, with the perennial fraction being mostly enclosed in the Arctic basin (Comiso, 2009). The ice cover in the Baltic Sea, however, has no perennial fraction and

can be highly variable due to the milder climate, often resulting in different freezing and melting periods during the same winter (Granskog et al., 2006). The sea ice season usually lasts for six to eight months, starting in October or November in the Bothnian Bay and the Gulf of Finland. Maximum extent is also normally reached in March (Haapala et al., 2015). Therefore, given the particular characteristics of the Baltic sea, the sea ice extent analysis is done by splitting the study area in two regions:

the NE Atlantic-Barents region (completely including the Iceland, Norwegian, Barents and White seas) and the Baltic region.

## 2.2   Selected data

Data used in this work consists of MODIS Terra level 1B Top-of-Atmosphere (ToA) radiance products MOD021KM (MODIS Science Team, 2017a), MOD02HKM (MODIS Science Team, 2017b) and the MOD35_L2 cloud mask (MODIS Atmosphere

Science Team, 2017), as summarized in Table 1. Swath data are resampled to 500 m resolution if necessary, converted to GeoTIFF format and projected to North Pole Lambert Azimuthal Equal Area using HDF-EOS To GeoTIFF Conversion Tool (HEG) v2.15 (NASA, 2019). No stitching is applied, as each scene is processed individually. However, scenes are clipped according to the selected study area. IceMap500 uses ToA radiance as input data which is later converted to ToA reflectance or ToA brightness temperature, so there is no atmospheric correction. Note that the objective of the algorithm is to map sea ice

presence rather than using reflectance as a proxy to get other physical variables such as sea ice concentration, so the absence of atmospheric correction reduces processing time, facilitates the algorithm's replicability and ensures the consistency of the dataset.

**Table 1.** MODIS Terra swath data used in this work. Accessible at the NASA's Level-1 and Atmosphere Archive (https://ladsweb.modaps. eosdis.nasa.gov.)

| Band | Bandwidth | Spectrum region | Code |
|---|---|---|---|
| **MOD02HKM (bands 1-7 at 500 m resolution)** | | | |
| 2 | 841-876 nm | Near-infrared (NIR) | B2 |
| 4 | 545-565 nm | Green (G) | B4 |
| 7 | 2.105-2.155 μm | Short-wavelength infrared (SWIR) | B7 |
| **MOD021KM (bands 8-36 at 1 km resolution)** | | | |
| 20 | 3.660-3.840 μm | Mid-wavelength infrared (MWIR) | B20 |
| 32 | 11.770-12.270 μm | Thermal infrared (TIR) | B32 |
| **MOD35_L2 (cloud mask product)** | | | |

## 2.3 Overview of previous MODIS sea ice extent algorithms

IceMap500 is fundamentally based on the previous IceMap (Riggs et al., 1999; Hall et al., 2001) and IceMap250 (Gignac et al., 2017) algorithms and inherits many of their features. Both algorithms feature a classification strategy based on threshold tests, but differ on the cloud masking approach. IceMap uses the Normalized Difference Snow Index (NDSI, Eq. 1) as the main criterion to classify sea ice, followed by a ToA threshold test using MODIS B4 (545-565 nm). To prevent misclassification of clouds as sea ice, this algorithm uses the MOD35_L2 cloud mask as an input, and outputs sea ice extent at 1 km resolution.

$$NDSI = \frac{B4 - B6}{B4 + B6} \tag{1}$$

Instead, IceMap250 uses the Normalized Difference Snow and Ice Index 2 (NDSII-2, Eq. 2) (Keshri et al., 2009), as well as the same ToA reflectance threshold at 545-565 nm to classify sea ice and water. The threshold value of the NDSII-2 is determined by splitting data in two groups with a Jenks natural breaks optimization (Jenks, 1967), which maximizes inter-class variance and minimizes intra-class variance. This algorithm features a hybrid cloud masking approach designed to minimize error while maximizing the mapped area, using the MOD35_L2 cloud mask alongside an additional visibility (VIS) mask, both at 1 km resolution.

$$NDSII2 = \frac{B4 - B2}{B4 + B2} \tag{2}$$

The VIS mask in IceMap250 is intended to identify areas where visibility is sufficient to perform a classification, for the sole goal of detecting open water. It uses the normalized difference between the MODIS thermal bands B20 and B32 as in Eq. 3.

$$R_{(B20/B32)} = \frac{B20 - B32}{B20 + B32} \tag{3}$$

The standard score of $R_{(B20/B32)}$ is then calculated, as seen in Eq. 4, where $\mu$ and $\sigma$ are the mean and standard deviation of $R_{(B20/B32)}$ of the swath data to be classified. Pixels where VIS < 0.5 are tagged as having enough visibility. The masking produces the MOD35 and the VIS datasets, which are classified separately and later combined following the set of rules in Table 2.

$$VIS = \frac{R_{(B20/B32)} - \mu}{\sigma} \tag{4}$$

Although masking in IceMap250 is done at a resolution of 1 km, the algorithm maps sea ice and water at 250 m within the masked area. This is accomplished by means of a downscaling technique by Trishchenko et al. (2006).

**Table 2.** IceMap250 possible combinations of the classified maps and corresponding outputs (Gignac et al., 2017).

| MOD35 map | VIS map | Composite map |
|-----------|---------|---------------|
| ice | ice | ice |
| ice | water | water |
| ice | NoData | NoData |
| water | ice | NoData |
| water | water | water |
| water | NoData | NoData |
| NoData | ice | NoData |
| NoData | water | water |

## 2.4 IceMap500: challenges and improvements

Both IceMap and IceMap250 face some challenging limitations which IceMap500 tries to address. The most important issue arises from the MOD35_L2 cloud mask, as it occasionally features resolution-breaking square artefacts of 25 km side length along the ice edge (Fig. 2) that prevent its accurate mapping. Such artefacts originate in the setting of the snow/ice background flag during the production process of the mask (Riggs and Hall, 2015), in which NSIDC's Near-real-time Ice and Snow Extent (NISE) product (Brodzik and Stewart, 2016), based on SSM/I-SSMIS passive microwave data with a cell size of 25 km, is used to determine the flag's state. Therefore, as the cloud detection algorithm takes different paths depending on the background flag, sea ice falling outside the footprint of the NISE classification is ultimately tagged as cloud in MOD35_L2. These 25 km artefacts can occupy extensive areas in some scenes, causing the loss of many cloud-free classifiable pixels.

Another notable source of classification errors, this time only in IceMap250, arises from the NDSII-2 test, which uses the Jenks natural breaks optimization to split pixels in two groups, regardless of the surface classes present in a scene. When batch processing MODIS data it may be likely to run into scenes lacking either ocean water or sea ice and, consequently, the Jenks optimization splits pixels into both surface classes erroneously. Clouds that are undetected by the MOD35_L2 cloud mask algorithm (Ackerman et al., 2010) and sun glint over ocean water may also be common error sources due to the similar reflectance characteristics to sea ice, both in IceMap and IceMap250. Additionally, as stated in Gignac et al. (2017), bare ice and melt ponds may also fail the classification tests due to the similarity with ocean water.

To mitigate those potential classification errors, IceMap500 features changes in the data masking and the classification rules. The new algorithm uses the dual masking approach and the NDSII-2 and B4 ToA reflectance tests as IceMap250, but increases the restrictiveness of the masking and the classification. It also introduces an additional Sea Surface Temperature (SST) test, and a new MOD35 correction workflow to minimize the effect of the NISE footprint and enlarge the mapped area (see the structure in Fig. 3). The downscaling technique used in IceMap250 is not applied for various reasons: I) simplicity, II) reduced

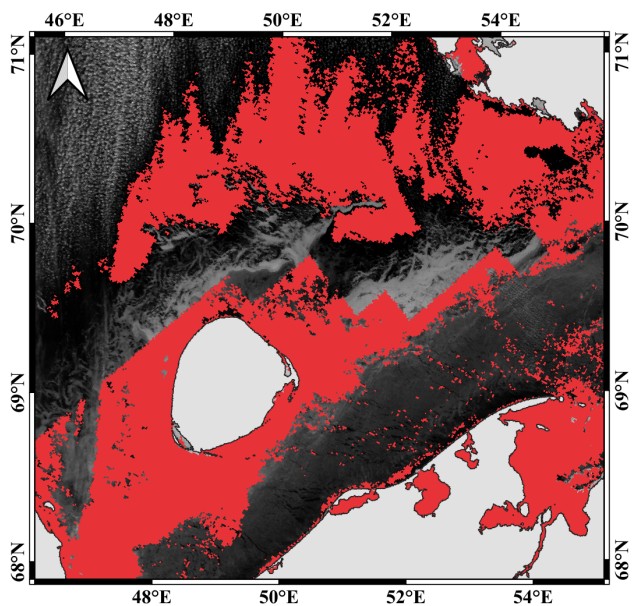

**Figure 2.** Pixels tagged as confident clear in the MOD35_L2 cloud mask, shown in red, overlaying MODIS B4 swath data from March 2012 (Barents sea). The footprint left by NISE on the cloud mask can be clearly seen along the ice edge.

processing times, III) MODIS Aqua band to band registration errors which may be even larger than the 250 m cell size itself (Xiong et al., 2006; Khlopenkov and Trishchenko, 2008), and IV) spectral integrity of the imagery (since no downscaling is applied). IceMap500 swath maps can be aggregated at any desired time scale. We use a map aggregation approach which is
sensitive to spatio-temporal variations of sea ice and which can be used to filter out unreliable sea ice classifications. The next sections give a more in-depth explanation of the IceMap500 workflow.

### 2.4.1   The masking

IceMap500 uses the same hybrid cloud masking approach as IceMap250. The VIS mask is used and calculated as in IceMap250, using the same VIS < 0.5 threshold value. Therefore, IceMap500 also generates the MOD35 and the VIS datasets. Nevertheless,
the MOD35 mask includes additional constraints so not only cloud cover is considered, but also the lighting conditions, sun glint and the presence of land. This information is contained within the MODIS product MOD35_L2, which provides multiple quality assessment flags (Strabala, 2004; Ackerman et al., 2010). We use the following flag states:

1. *Unobstructed FOV*, selecting only pixels identified as confident clear, with a confidence of 99 % (Ackerman et al., 2010).

2. *Day/Night*, selecting only pixels identified as day. This flag is of special importance during the winter months, when the
polar twilight zone reaches the lowest latitudes and, therefore, the available daytime area becomes scarcer.

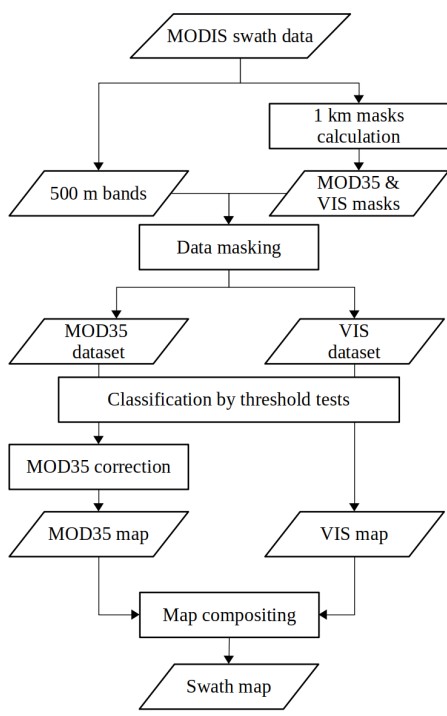

**Figure 3.** Simplified structure of IceMap500.

3. *Sun glint*, selecting only pixels identified as no sun glint. This way, areas with sun glint caused by the reflection angle of the sun being between 0° and 36° are discarded. Other potential sun glint sources are not considered (Ackerman et al., 2010).

4. *Land/Water*, selecting only pixels identified as water. Land masking is crucial to ensure the quality of the resulting classification because, as already pointed out in Gignac et al. (2017), an incorrect masking may generate sea ice false positives due to the reflectance contrast of land with water.

### 2.4.2 The classification tests

In IceMap500 three different threshold tests are included:

1. NDSII-2 test ($t_{ndsii2}$). Same as in IceMap250. The threshold value $k$ is determined by slicing the NDSII-2 (Eq. 2) with the Jenks natural breaks optimization. Pixels in the first group (i.e. NDSII-2$< k$) are classified as sea ice. This test was shown to discriminate 96-100 % of sea ice even during the melting periods in Gignac et al. (2017).

2. Green ToA reflectance test ($t_{b4}$). Same as in both IceMap and IceMap250. A pixel is tagged as sea ice if its reflectance is $\geqslant$17 % at 545-565 nm (B4). This threshold is based on the contrast in reflectance between ice and water at visible wavelengths, and was first used in Riggs et al. (1999) and later validated in Gignac et al. (2017). Gignac et al. (2017)

demonstrated that a B4⩾17 % threshold resides slightly into the upper standard deviation of the water class reflectance, so the risk of misclassifying melt ponds, leads, polynyas and low-albedo sea ice is low.

3. Mid-range infrared temperature test ($t_{b20}$). This new threshold is based on the Sea Surface Temperature (SST) using B20 (3.660-3.840 μm). It is always used in conjunction with $t_{b4}$, although only in the MOD35 dataset classification. Therefore, sea ice is classified only when both B4 ⩾ 17 % and SST < 1 °C. The $t_{b20}$ test is used as a sort of mask to
confirm that a pixel tagged as sea ice really belongs to sea ice, as unmasked sun glint, turbid water and aerosols may raise water reflectance past the $t_{b4}$ threshold. To perform this test B20 is temporarily atmospherically corrected with a straightforward equation used in the MODIS SST algorithm (Brown and Minnett, 1999) for mid-range infrared SST derivation (Eq. 5):

$$SST = 1.01342 + 1.04948 T_{B20} \tag{5}$$

where $T_{B20}$ is the brightness temperature of B20. The 1 °C threshold is designed to include leads, cold water, new sea ice and melt ponds (which according to Zhang et al. (2017) typically stay below 0.3 °C) to prevent breaking the 500 m resolution, while still discarding most open water (refer, for instance, to global SST products such as NOAA High Resolution SST by NOAA/OAR/ESRL PSL, Boulder, Colorado, USA, available at https://psl.noaa.gov/data/gridded/data. noaa.oisst.v2.highres.html). Moreover, B20 may be contaminated by reflected solar radiation, causing $T_{B20}$ to increase
and therefore making easier the exclusion of sun glint.

In addition, IceMap500 features restrictive classification rules to compensate the output of $t_{ndsii2}$ in scenes with a single surface class, as the Jenks optimization will still split data in two groups. The classification rules depend on the dataset that is being classified, as when merging the MOD35 and VIS maps changes in a single dataset classification ultimately affect the whole outcome. The classification rules are shown in Table 3: sea ice is only mapped in the MOD35 dataset when there is
consensus between the tests, while in the VIS dataset it is mapped whenever $t_{b4}$ is positive. A downside of this method is that it may leave some melt ponds as NoData, since in the most advanced melting states they tend to show NDSII-2 values similar to water (Gignac et al., 2017). Note that while masking is done at 1 km resolution, the swath data that is classified is at 500 m, so sea ice and water are mapped at 500 m within the mask limits.

### 2.4.3  MOD35 correction

Once the MOD35 map is created, an additional set of tests is introduced to attenuate the effects of the NISE footprint present in the MOD35_L2 mask, which propagate to the MOD35 classification and ultimately to the composite maps. Although the inclusion of this correction increases the chances of classification errors, it greatly improves the sea ice edge delineation and increases the classified area. The MOD35 correction is designed to reclassify NoData pixels within a buffer zone surrounding

**Table 3.** Classification outcomes based on the threshold tests in IceMap500.

| MOD35 dataset | | | VIS dataset | | |
|---|---|---|---|---|---|
| $t_{ndsii2} < k$ | $t_{b4} \geqslant 17\,\%$ $t_{b20} < 1\,°C$ | MOD35 map | $t_{ndsii2} < k$ | $t_{b4} \geqslant 17\,\%$ | VIS map |
| yes | yes | ice | yes | yes | ice |
| yes | no | NoData | yes | no | NoData |
| no | yes | NoData | no | yes | ice |
| no | no | water | no | no | water |

clusters of sea ice. Within this buffer the MOD35_L2 cloud mask is ignored during the classification. Instead, MODIS B7 (2.105-2.155 µm) is used to detect clear areas by taking advantage of the very low reflectance values that water, snow and ice display at such wavelengths, allowing cloud discrimination (e.g. Platnick et al., 2001; Thompson et al., 2015). To avoid error amplification, sea ice clusters below 100 pixels are deleted before the correction: if those clusters are found far from the ice edge it is likely that they originate from sun glint or unmasked clouds, while those found close to large clusters of sea ice are ultimately classified again as such. The MOD35 correction includes five tests, as illustrated in Fig. 4.

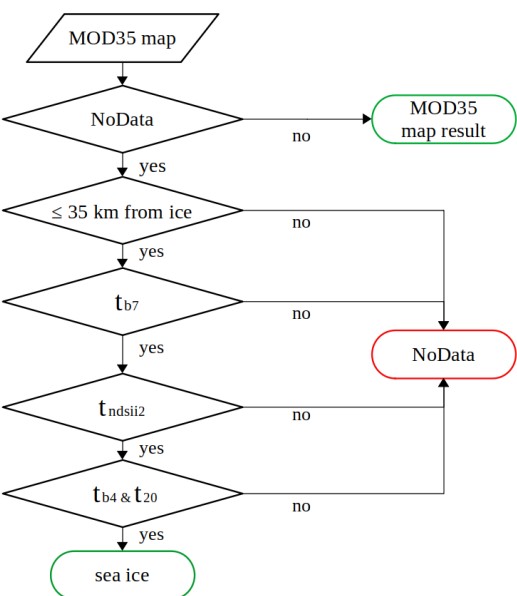

**Figure 4.** MOD35 block correction structure and possible test outcomes.

1. NoData test. NoData pixels pass the test, while classified areas remain the same. All pixels set as NoData during the MOD35 classification also undergo the tests, and may be finally labelled as sea ice or water.

2. Euclidean distance test. NoData pixels found at 35 km or closer to a cluster of sea ice pass the test; otherwise are set as NoData. This distance is roughly equal to the diagonal of NISE's 25 km cells, and is used to reduce the chances of misclassifying clouds as sea ice by setting a buffer along the ice edge.

3. Short-wavelength infrared ToA reflectance test ($t_{b7}$). Pixels below 3.5 % ToA reflectance at 2.105-2.155 μm (B7) pass the test, otherwise are set as NoData. This threshold is based on the low reflectance that water, snow, and ice display around 2 μm: spectral signatures in Fig. 5 indicate a maximum reflectance of $\sim$10 % for ice within the B7 bandwidth, while the reflectance of snow and water is always below 5 %. This test is used as a cloud filter, as it is expected that clouds show higher reflectance values. Fig. 5 also shows the threshold includes only 45.3 % of clear areas according to our sampling, although most of the remaining samples belong to sea ice far from the ice edge which is of no interest in the MOD35 correction. However, by setting such a restrictive threshold only a small fraction of clouds are included (1.5 %), which is preferred over including all sea ice while increasing significantly sea ice false positives due to the cloud cover.

4. $t_{ndsii2}$. Same as in the MOD35 classification. Pixels where NDSII-2$< k$ pass the test, otherwise are set as NoData. In this case, the Jenks optimisation is not performed using all the clear pixels in the scene, but rather only those within the 35 km buffer zone set as clear by $t_{b7}$.

5. $t_{b4}$ & $t_{b20}$. Same as in the MOD35 classification. Pixels where B4 $\geqslant 17$ % and SST $< 1$ °C are classified as sea ice, otherwise are set as NoData.

Finally, the MOD35 map and the result of the MOD35 block correction are merged and later combined with the VIS map according to the compositing rules in Table 2. A visual example of the workflow in IceMap500 is given in Fig. 6, illustrating each intermediate result of the algorithm.

### 2.4.4 Map aggregation and calculation of sea ice extent

The corrected MOD35 and VIS maps created for each scene are combined to take advantage of the strengths of both the MOD35 and the VIS classification methods, following the criteria seen in Table 2. The extensive cloud cover found in most scenes and the restrictiveness of the classification implies that only a small area is finally mapped, although the new correction reduces the impact of the cloud mask. In any case, many scenes are required to map large expanses of the sea ice cover. In IceMap500 a map aggregation method based on the number of coincident sea ice classifications achieved in each pixel is used, meaning that pixels classified as sea ice in a large number of scenes will have higher reliability. The aggregated maps are generated by calculating the sum of composite maps where ice = 1 and water = 0, and later normalizing the results according to the maximum number of coincident sea ice observations achieved. With MODIS Terra the maximum number of observations

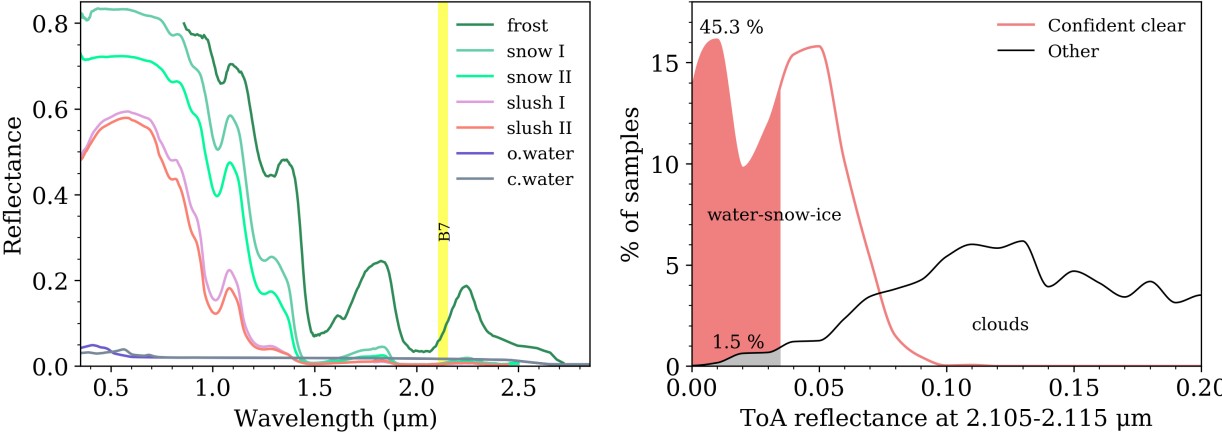

**Figure 5.** Left: spectral signatures of several surfaces obtained from the USGS spectral library (Kokaly et al., 2017), including ice (frost), sea water (oceanic and coastal), and snow-slush at different melting states (indicated by roman numerals); MODIS band 7 bandwidth is shown in yellow. Right: histograms for pixels identified as confident clear and other (probably clear, uncertain clear, and cloudy) in the MOD35_L2 product, from 8000 randomly sampled points on five different scenes. Percentages indicate the proportion of pixels inside each filled area using a 3.5 % ToA reflectance threshold.

typically reaches ∼50 in March and ∼60 in September, so using both Terra and Aqua this number could double and signifi-
235 cantly increase the usefulness of this method. The output provides information about where is sea ice more likely to be found,
thus we appropriately refer to the resulting maps as sea ice presence likelihood maps (Fig. 7). This approach allows users to
detect the places where sea ice has been more unstable during a given time period, as the sea ice presence likelihood will drop
in such cases. Likelihood maps allow even to detect cracks in the sea ice, and of course if sea ice has moved significantly the
sea ice presence likelihood will be lower.

Sea ice extent is obtained from the likelihood maps by selecting a likelihood threshold, in this case 10 %. Then, pixels where
sea ice presence is >0 % and <10 % (0 % is water) are discarded because such observations might not be reliable enough. By
eliminating such observations a small NoData buffer zone along the ice edge is generated. IceMap500 then takes advantage
of the pixels set as water and fills the NoData gaps using an Euclidean distance allocation method. This way a clearer and
245 smoother sea ice edge is obtained, which nonetheless does not ignore the information carried by pixels where likelihood falls
below the selected threshold. This procedure acts as an additional post-classification error filter and produces a sea ice extent
map, as the constant motion of the ice tends to hide the presence of features such as leads, cracks, polynyas and ice floes.

It is worth noting that the aggregation procedure eventually sets either as sea ice or water NoData pixels that were never re-
250 ally classified by the algorithm during the entire time period. Although the dual masking approach and the use of the MOD35
correction greatly improve the final classified area, NoData gaps still tend to appear in the regions closer to the pole in the

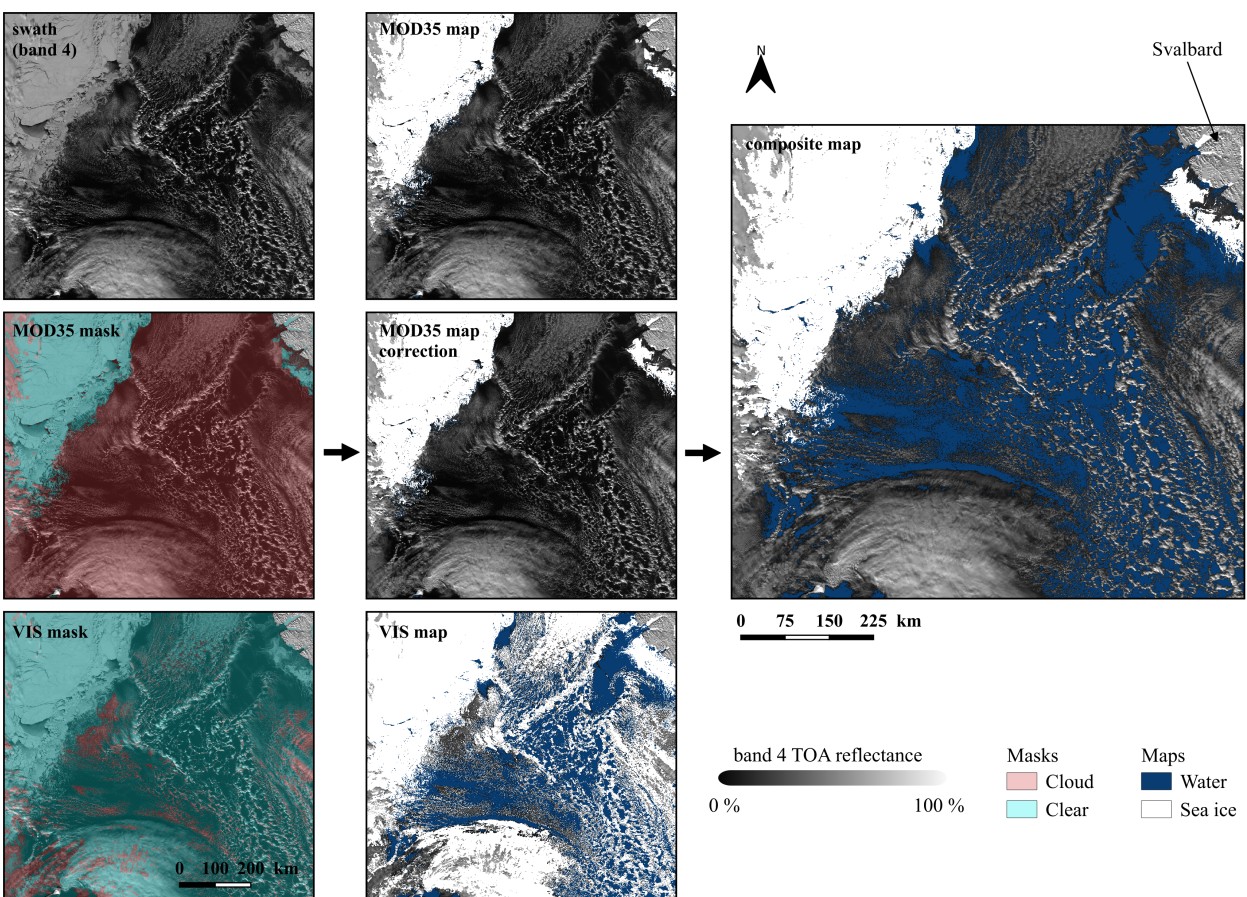

**Figure 6.** Intermediate and final products of IceMap500. The effect of the MOD35 correction is best seen on the upper right corner of the maps.

March monthly maps as a consequence of the poor lighting conditions. This also makes sea ice presence likelihood to drop. September has no such lighting limitations, so NoData gaps appear more randomly. Fortunately, the average NoData area fraction of our monthly time series only reaches 1.0 % in March and 0.7 % in September.

## 3 Results

We use the new IceMap500 algorithm to obtain swath and daily maps during the months of March and September of the 2000-2019 period, using only MODIS Terra data. The resulting maps have been aggregated at a monthly scale to obtain the time series of sea ice extent, from which sea ice extent trends have been calculated. The performance of the algorithm is assessed
with confusion matrices by manually validating swath maps.

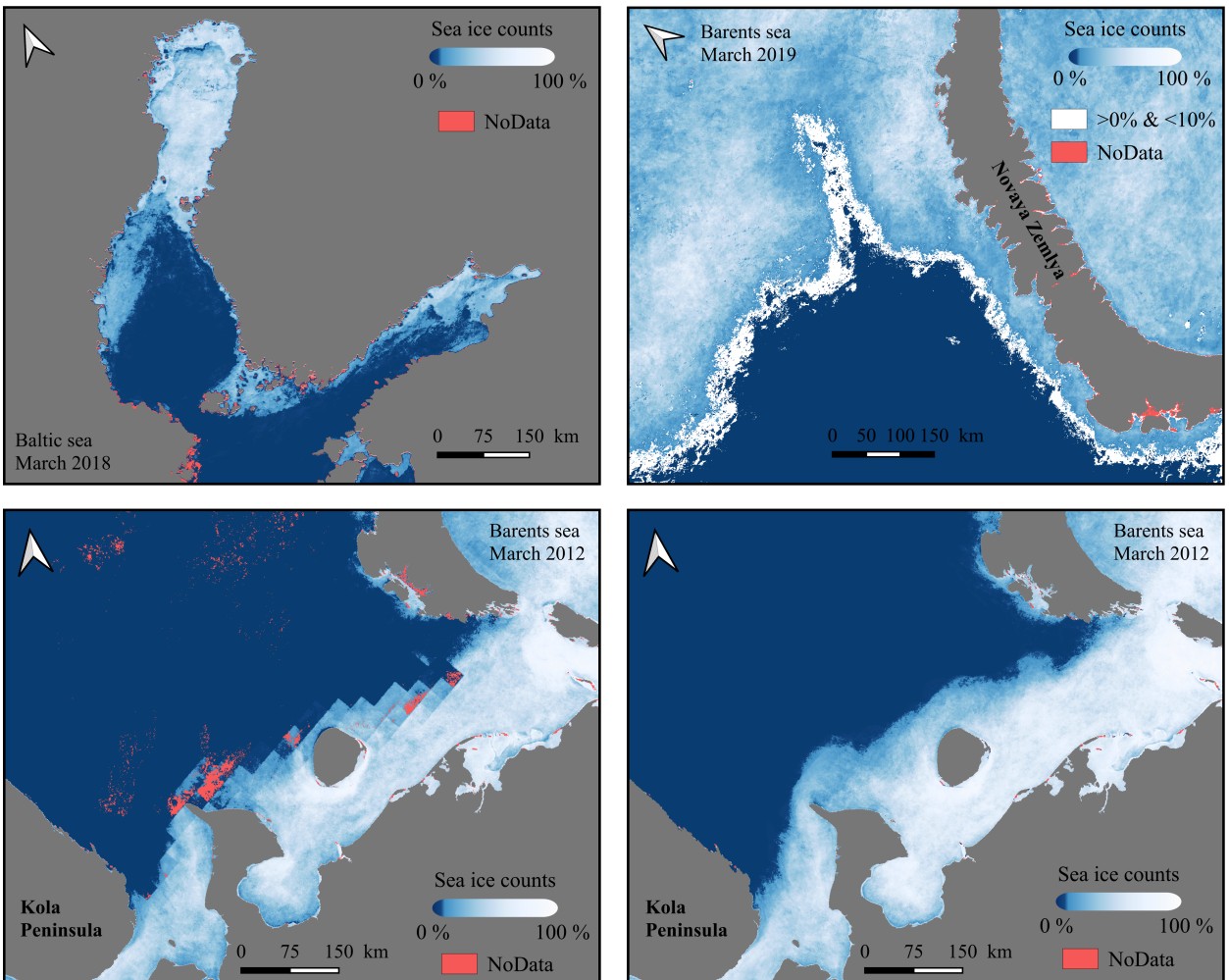

**Figure 7.** Clockwise from upper left: example of monthly sea ice presence likelihood in the Baltic sea; buffer zone generated when setting a 10 % likelihood threshold; monthly sea ice likelihood without MOD35 correction; monthly sea ice likelihood with MOD35 correction.

## 3.1 Sea ice extent evolution and trends

Monthly sea ice extent maps have been used to determine the sea ice extent trends between 2000 and 2019 in the NE Atlantic-Barents region and the Baltic Sea separately. Both March and September trends have been obtained for the NE Atlantic-Barents, that is, the trends of the maximum and minimum sea ice cover, respectively. Since there is no perennial sea ice fraction in the Baltic Sea, only the March trend is available in this case, also corresponding to the maximum sea ice cover. The resulting trend lines, represented in Fig. 8, have been obtained via least-squares linear regression.

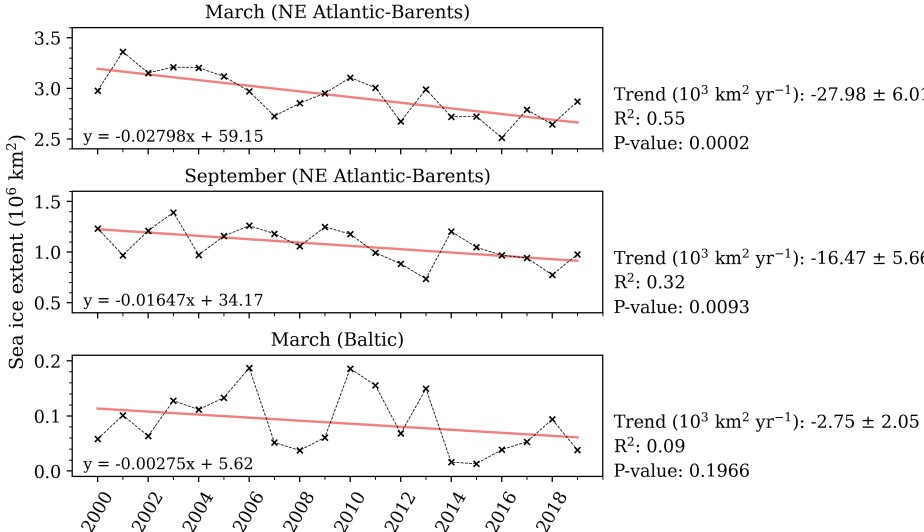

**Figure 8.** Monthly sea ice extent evolution and trend lines, alongside the numerical results of the sea ice trends and the standard error of the slope. Two goodness of fit estimators are given: the coefficient of determination and the p-value. P-values are obtained from two-tailed Wald tests with 18 degrees of freedom and null hypothesis that there is no correlation between the two variables, i.e. that the slope of the trend line is zero.

Results indicate that in the NE Atlantic-Barents region the sea ice decline is ~70 % faster in March than in September. Although September's extent is comparatively smaller, the standard error of the trends is similar in both months (~6×10$^3$ km$^2$yr$^{-1}$), with R$^2$ being lower in September. Nevertheless, both trends have been found to be statistically significant when considering a significance level of 99 %. In particular, the March trend displays a very low p-value, indicating a significance level of ~99.98 %. In contrast, the Baltic Sea displays no clear tendency and a large variability. This causes R$^2$ to be very low and the standard error of the trend to be almost equal to the trend itself. If the 99 % significance level criterion is followed then $H_0$ can not be rejected in the Baltic sea.

### 3.2 Accuracy assessment

We randomly selected eight years to perform the quality assessment, from which a total number of 32 scenes have been used, that is, two scenes per month to allow comparison. As a prerequisite, each validation scene must have both sea ice and water pixels. Validation has been carried out with confusion matrices by generating 1500 random points per scene over the classified areas. Those points have been manually tagged as either sea ice, water or cloud, with the help of the corresponding RGB swath. Although no clouds are mapped in the algorithm, points found over clouds opaque enough to avoid the identification of the surface below add to the total sea ice commission error.

Accuracy assessment results are summarized in Table 4. All scenes achieved overall accuracies above 90 %, resulting in an average accuracy of 96.0 %. The average kappa coefficient of 0.85 indicates a strong agreement between classification and ground truth, despite being affected by scenes with few water validation points, causing the kappa coefficient to drop due to the disproportion between classes. Individually, only 5 out of 32 computed kappa coefficients are found below the 0.80 value, while 10 are found between 0.80-0.90 and 17 above 0.90, indicating very strong agreement. The primary source of error affecting the classification is sea ice commission, with its mean value alone adding up to 7.3 %, that is, more than sea ice omission, water commission, and water omission combined.

By analysing separately both months, mean accuracy is found to be higher in March than September, differing by 1.9 %. Accuracy results in September are also slightly more variable, showing a $\sigma$ of 2.8 % versus 2.5 % in March. On the contrary, the mean kappa coefficient is lower in March than in September. This is linked to the much greater sea ice area covered in March, which occasionally causes some scenes to have very few water validation points, making the kappa coefficient to drop due to the disparity in validation points between classes. The standard deviation of kappa greatly illustrates this issue, being 0.23 in March and 0.06 in September.

Nevertheless, the difference in accuracy between months does not arise from validation artefacts, but mainly from the disparity in sea ice commission. With a mean sea ice commission error of 2.5 %, March classifications outperform those for September, which show a mean error of 12.2 %. Since there are only two classes, high water omission error should be expected. However, it is very low in both cases, 0.3 % in March and 0.04 % in September, revealing the dominance of sea ice commission is not caused by the misclassification of water as sea ice, but of clouds as sea ice. Instead, sea ice omission error is similar in both months, being 2.7 % in March and 3.3 % in September, while water commission is 2.5 % and 1.9 %, respectively. Thus, globally, the major error contribution is due to the misclassification of clouds as sea ice, especially in September, while misclassification of sea ice as water and water as sea ice remain lower in the first case and minimal in the latter.

According to Chan and Comiso (2013), the MOD35_L2 cloud mask tends to underestimate the cloud cover over sea ice, whereas over open water it is overestimated but closer to reality. Indeed, most sea ice commission error in our validation is due to the misclassification of clouds as sea ice within the limits of the sea ice cover; in fact, despite the cloud fraction being much larger over open ocean than over sea ice, in the first case sea ice commission errors are uncommon. Some of the clouds that are commonly left undetected by the MOD35 cloud mask include low-level (top below 2 km), high-level (top above 6 km), and thin clouds less than 2 km thick (Chan and Comiso, 2013). Additionally, our validation showed that multilayered clouds cast shadows which can be finally tagged as sea ice. The rise of sea ice commission error during September may be explained by the fact that, as shown by Chan and Comiso (2013), late summer in the Arctic is considerably cloudier than winter, as lower sea ice concentration relates to a larger cloud fraction.

**Table 4.** Validation results for 32 swath maps. Two results are given per month, corresponding to different scenes. Commission (com.) and omission (om.) errors represent the monthly mean. Kappa coefficients corresponding to scenes in which water validation points are less than 5 % from the total are shown in italics. The kappa statistic rates the agreement between classification and ground truth, although considering that agreement may occur by chance (Cohen, 1960).

| Year | Accuracy (%) | | Kappa coefficient | | Sea ice com./om. (%) | | Water com./om. (%) | |
|---|---|---|---|---|---|---|---|---|
| | March | September | March | September | March | September | March | September |
| 2003 | 99.3, 97.9 | 94.0, 91.1 | *0.66*, 0.88 | 0.88, 0.84 | 0.7 / 9.5 | 16.4 / 0.0 | 1.1 / 0.0 | 0.0 / 0.0 |
| 2005 | 95.3, 92.7 | 95.5, 99.1 | 0.95, 0.93 | 0.88, 0.97 | 0.0 / 6.2 | 10.2 / 2.1 | 5.4 / 0.0 | 0.4 / 0.0 |
| 2006 | 98.1, 98.6 | 94.1, 92.1 | 0.96, 0.97 | 0.88, 0.82 | 2.5 / 1.6 | 11.5 / 9.0 | 2.4 / 0.1 | 5.6 / 0.1 |
| 2008 | 97.4, 97.9 | 95.8, 98.0 | 0.94, 0.90 | 0.88, 0.95 | 2.2 / 1.0 | 11.6 / 1.3 | 2.8 / 0.0 | 0.4 / 0.0 |
| 2010 | 91.7, 97.8 | 90.8, 91.5 | *0.32*, 0.96 | 0.83, 0.78 | 5.5 / 0.6 | 19.6 / 9.5 | 0.9 / 0.4 | 3.5 / 0.1 |
| 2011 | 98.3, 98.3 | 95.7, 98.6 | 0.96, 0.96 | 0.91, 0.97 | 0.4 / 2.0 | 3.9 / 2.1 | 4.8 / 0.1 | 2.9 / 0.1 |
| 2014 | 98.7, 99.1 | 92.4, 94.7 | 0.93, 0.98 | 0.84, 0.85 | 1.0 / 0.9 | 22.2 / 0.9 | 1.9 / 0.0 | 0.3 / 0.0 |
| 2016 | 93.7, 91.2 | 97.0, 99.2 | *0.32, 0.50* | 0.94, 0.98 | 7.8 / 0.0 | 1.8 / 1.5 | 0.7 / 2.1 | 1.9 / 0.0 |
| **Mean** | **96.9** | **95.0** | **0.82** | **0.89** | **2.5 / 2.7** | **12.2 / 3.3** | **2.5 / 0.3** | **1.9 / 0.0** |
| **Total** | | | | | | | | |
| **Mean** | **96.0** | | **0.85** | | **7.3 / 3.0** | | **2.1 / 0.2** | |
| **Median** | **97.2** | | **0.91** | | **4.7 / 1.5** | | **1.9 / 0.0** | |

Since sun glint issues have been mostly solved, as evidenced by the minimal impact of water omission error, and most sea ice commission is generated within the sea ice cover, there are few clusters of sea ice false positives over open ocean, most of which are removed during the MOD35 block correction. Thus, few of those errors are propagated to the sea ice presence likelihood maps, allowing the selection of low threshold values to obtain sea ice extent.

### 3.2.1 Agreement with NSIDC's Sea Ice Index

The Sea Ice Index (SII, Fetterer et al., 2017) is a widely used global sea ice extent and concentration product distributed by the NSIDC, which is derived from satellite passive microwave data at 25 km spatial resolution. It covers from 1978 to the present, being updated on a daily basis, and provides monthly median sea ice extent maps. In the SSI, extent is derived from sea ice concentration by setting as sea ice pixels where concentration is 15 % or above. In spite of the difference in spatial resolution between the SII and IceMap500, measuring the agreement or similarity between both datasets can act as an estimator of the quality and consistency of IceMap500's monthly aggregates. Thus, SII maps have been reprojected to North Pole Lambert Azimuthal Equal Area and resampled down to a 500 m cell size. Then agreement has been calculated as the coincident sea

ice area fraction between both datasets, as compared to the total sea ice extent including coincident and non-coincident area (Eq. 6).

$$Agreement = \frac{A \bigcap B}{A \bigcup B}$$ (6)

where $A$ is an IceMap500 monthly aggregate and $B$ the corresponding SII. Fig. 9 illustrates the agreement both for March and September from 2000 to 2019. Mean agreement in March is 89.5 % with a standard deviation of 1.1 %, whereas in Septem-

ber mean agreement is lower, 85.5 %, and displays higher variability, with a standard deviation of 3.1 %. Only in a single case does the agreement fall below 80 %, corresponding to September 2013 (74.7 %).

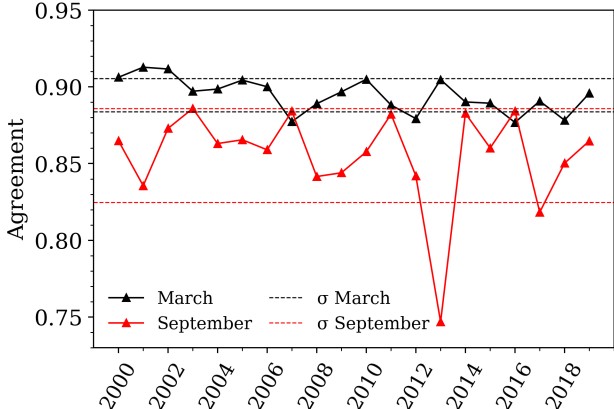

**Figure 9.** Agreement between NSIDC's Sea Ice Index and the obtained monthly sea ice extent maps for all analysed years.

   An example of both datasets is shown in Fig. 10 for visual comparison: even though the difference in spatial resolution is not compensated, both numerical and visual analysis suggest that IceMap500 monthly aggregates are coherent with existing

data even considering the different sea ice extent calculation approach.

## 4   Discussion

### 4.1   Sea ice trends

Sea ice trends obtained from our monthly extent maps in the NE Atlantic-Barents region are consistent with previous observations and both are statistically significant when considering a significance level of 99 %. The trends obtained in this study are

regional and therefore do not reflect the overall Arctic sea ice extent tendencies, although they can be compared to studies in which regional trends are also analysed. In Cavalieri and Parkinson (2012) the summation of sea ice trends (1979-2010) in the Greenland sea and the Barents-Kara seas, roughly corresponding to our study area, shows a greater loss of sea ice extent during

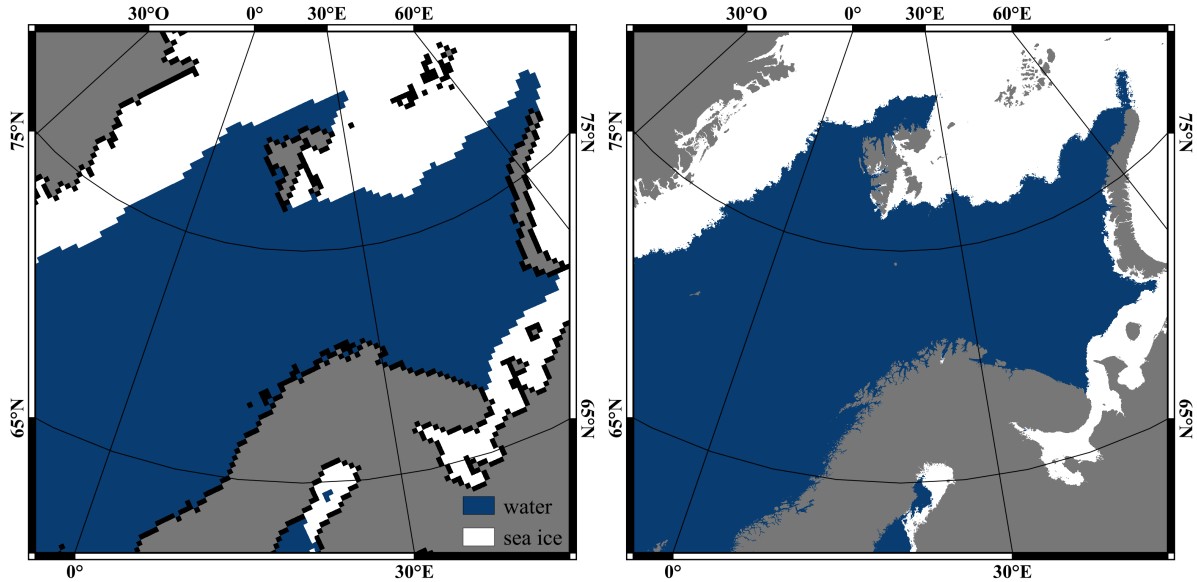

**Figure 10.** Comparison between NSIDC's Sea Ice Index (left) and sea ice extent map obtained for March 2012 (right).

winter ($-21.7\pm3.1\times10^3$ km$^2$yr$^{-1}$) than during summer ($-18.6\pm3.2\times10^3$ km$^2$yr$^{-1}$). This is in accordance with our results, and both trends are within the error range of the trend lines in Fig. 8. Similarly, the summation of the Greenland-Barents-Kara

trends in Peng and Meier (2017), covering from 1979 to 2015, indicates a trend of $-19.0\pm4.4\times10^3$ km$^2$yr$^{-1}$ for the maximum sea ice extent and $-14.9\pm5.7\times10^3$ km$^2$yr$^{-1}$ for the minimum. This behaviour is also reported in the Barents sea in Kumar et al. (2021), spanning the 1979-2018 period. Nevertheless, the sea ice extent loss is proportionally smaller in winter than in summer: in our study area the decadal sea ice loss is approximately of 9 % in March and 13 % in September. Peng and Meier (2017) report sea ice losses of 10.1 % and 10.8 % per decade in the Greenland and Barents seas in winter, closely matching our results.


In the case of the Baltic Sea, no statistically significant trend can be inferred due to high interannual variability and the limited lifespan of MODIS. This, however, does not imply that $H_0$ (i.e. that the Baltic ice cover is stable) is true: previous research (Jevrejeva et al., 2004) based on data from coastal observatories covering years 1900 to 2000 reveals a significant decreasing trend in sea ice occurrence probability in the southern Baltic Sea, while in the northern half ice occurs every winter.

Moreover, it shows a shortening of the sea ice season and an advance in the date of break-up, especially in the northern areas. More recent analyses (Vihma and Haapala, 2009; Haapala et al., 2015) also indicate that over the last century the sea ice season has shortened and the occurrence of severe winters has fallen. Thus, although IceMap500 may not be suitable for Baltic sea ice monitoring at a monthly scale due to the large variability, both interannual and within a same freezing period (Granskog et al., 2006), it can be useful for detailed sea ice studies spanning shorter time periods.


## 4.2 Applicability of IceMap500

Accuracy assessment shows that the major source of error in IceMap500 is sea ice commission, mostly caused by undetected clouds. This is especially true in September due to the cloudier atmospheric conditions during the Arctic summer. This issue is also reflected in the agreement with NSIDC's SII, with the September agreement being lower than in March in all but two years and occasionally falling down to 75 % (September 2013). The larger number of scenes available during that month alongside the larger sea ice commission error make the September monthly aggregates to be potentially affected by sea ice false positives to a greater extent, so a possible way of dealing with this situation is to increase the sea ice presence likelihood threshold. In the case of September 2013, a small change in the threshold value (from 10 % to 11 %) translates into an increase in IceMap500-SII agreement from 75 % to 80 % (see Fig. 11 for visual comparison). An additional source of disagreement between SII and IceMap500 in the summer months is the greater fragmentation of the ice cover, leading to the formation of sea ice floes, alongside the coastline discrepancy. However, these two sources are intrinsically linked to the difference in spatial resolution between both products. In the case of September 2013, the fragmentation of sea ice (notice the water pixels within the SII edge in Fig. 11) in combination with high sea ice commission due to unmasked clouds led to an unusually low agreement score.

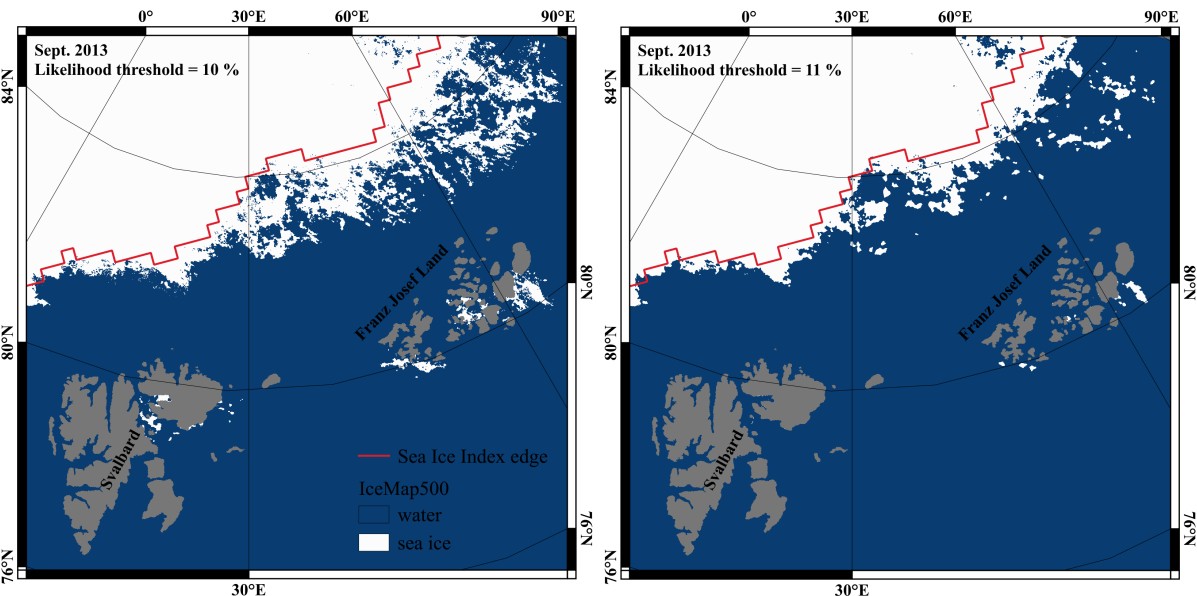

**Figure 11.** Comparison between IceMap500 and the SII (September 2013) using two sea ice presence likelihood thresholds: 10 % (left panel) and 11 % (right panel).

Overall, both the accuracy assessment of IceMap500 and the generally high agreement values with the SII suggest that the new algorithm is well suited for sea ice studies and monitoring. Its processing time also allows near real-time mapping: it takes

around 50 minutes to generate a full daily map covering our study area (i.e. 16 scenes) using a modest machine with an Intel Xeon X5550 (4x2.67 GHz) processor and 12 GB RAM. Therefore, IceMap500 may represent an improvement towards local and regional sea ice studies, especially taking into account the spatio-temporal information carried by the sea ice presence likelihood maps. Additionally, the inclusion of the MOD35 correction allows IceMap500 to map the sea ice edge more accurately than the MOD29 product (see the comparison in Fig. 12), which is visibly affected by the NISE footprint. This increase in mapped area is also advantageous when aggregating maps at any time scale, as sea ice presence likelihood rises and the presence of NoData gaps is minimized. Instead, in Fig. 12 the IceMap500 result is closer both in terms of mapped area and spatial resolution to the VIIRS/NPP sea ice cover (375 m) swath product (Tschudi et al., 2017). It is worth noting, however, that VIIRS products may also be affected by the VIIRS cloud mask in the same way that MODIS is, because NISE is also used to detect background sea ice in the VIIRS cloud mask algorithm (Frey et al., 2019).

Even though IceMap500 is designed to work with MODIS, it could also be used with other optical and infrared sensors, as long as the selected sensor has equivalent bands to those used by this algorithm. Nevertheless, the application of the MOD35 correction, which would have to be adapted, depends on the characteristics of the cloud mask to be used, and may not even be necessary. In the case of VIIRS the MOD35 correction may be advantageous due to the potential effect of the NISE background, but there is no direct equivalent to MODIS B7 which is used to identify clouds during the correction. Therefore, the potential of the closest match (VIIRS band M11, with a 2.20-2.30 μm bandwidth) to discern clouds from the ice cover should be assessed in this context. However, the application of the IceMap500 algorithm both to other sensors or other study regions might yield different accuracy assessment results, so the threshold tests or the classification restrictiveness might need to be revised in each particular case to improve its performance.

## 5 Conclusions

The new IceMap500 algorithm is shown to generate high quality sea ice maps by systematically achieving accuracies above 90 %. Quality assessment revealed the most common error is sea ice commission caused by unmasked clouds, manifesting the key role that cloud masking plays on the overall accuracy of the algorithm. The addition of the MOD35 correction substantially improves the delineation of the ice edge, preventing the propagation of the NISE footprint, and increases the mapped area, which is of capital importance when deriving daily and monthly maps due to the restrictiveness of the classification and the weather dependence of MODIS visible and infrared data. High agreement between our monthly sea ice extent maps and NSIDC's Sea Ice Index, especially in March, demonstrates the consistency of the map aggregation method and further exemplifies the overall good performance of the algorithm. Data produced by IceMap500 has proved useful to evaluate sea ice extent trends in the NE Atlantic-Barents region and the Baltic Sea. Significant negative trends have been observed both in March and September in the NE Atlantic-Barents region, while the Baltic Sea displays much more variability and no trend can be inferred from it. Given the high accuracies achieved and the coherence with existing data, we find that IceMap500 is a useful tool for sea ice studies and monitoring, particularly at local and regional scales.

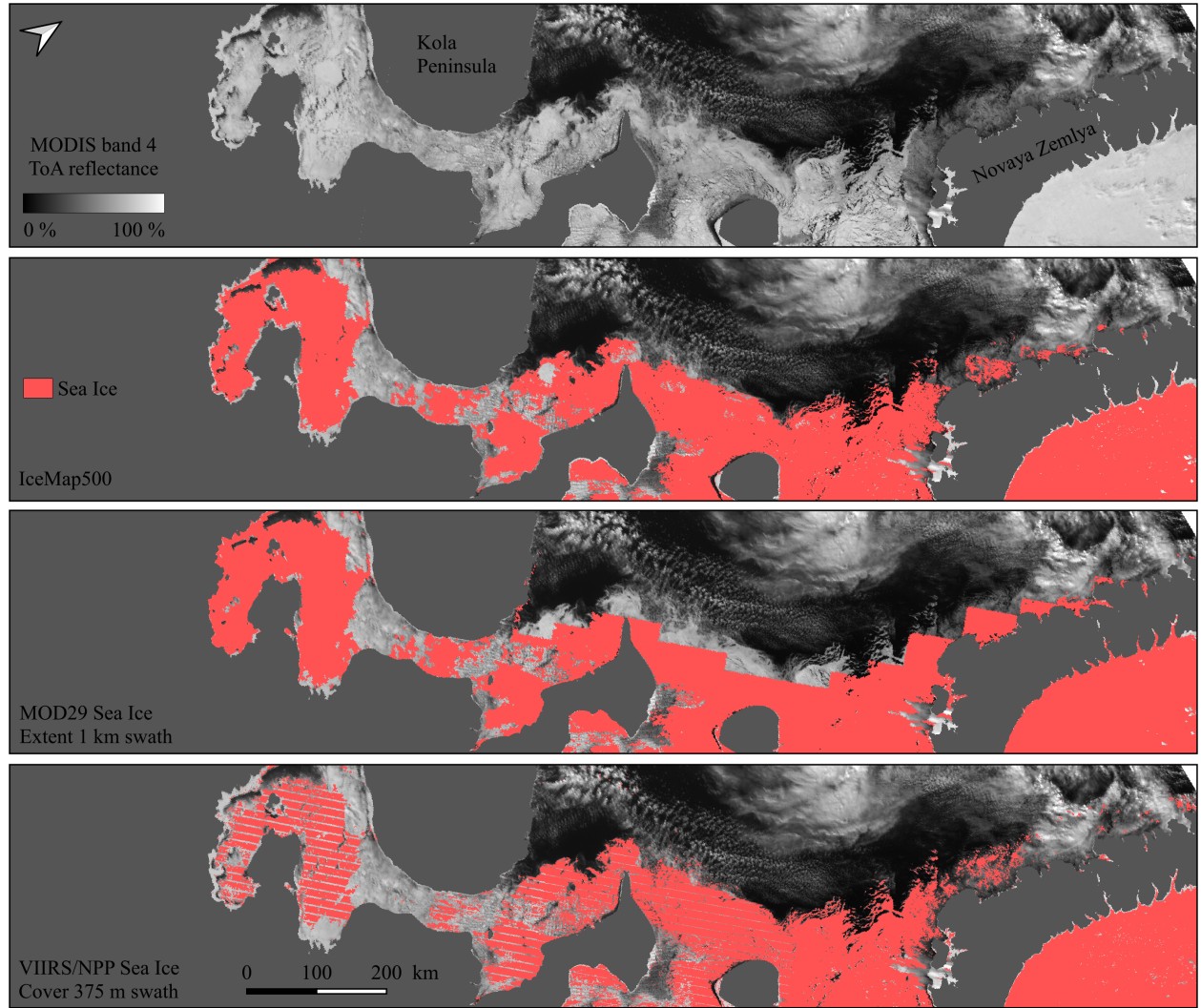

**Figure 12.** Comparison between IceMap500 swath composite, MOD29 sea ice extent and VIIRS/NPP sea ice cover products (March 26th 2018). For MODIS Terra the swath acquisition time is 7:40 UTC, for VIIRS/NPP it is 7:18 UTC. Disagreement between IceMap500 and MOD29 along the shoreline is attributed to land masking differences.

*Code and data availability.*   The source code is hosted at https://github.com/Parera-Portell/IceMap500. Monthly March and September sea ice extent maps from 2000 to 2019 are available at https://doi.org/10.5565/ddd.uab.cat/233396.

*Author contributions.*   Joan A. Parera-Portell: investigation, methodology, formal analysis, software, writing-original draft; Raquel Ubach: conceptualization, supervision, resources, writing-review and editing; Charles Gignac: supervision, writing-review and editing.

*Competing interests.* The authors declare they have no competing interests.

*Acknowledgements.* The authors thank two anonymous referees and the editor M. Sandells for their useful comments, which greatly improved the manuscript. Work on software QGIS and HDF-EOS To GeoTIFF Conversion Tool (HEG) is acknowledged. Special thanks to Jaume Fons-Esteve at the Department of Geography, Universitat Autònoma de Barcelona.

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
