# Peer review of "An improved sea ice detection algorithm using MODIS: application as a new European sea ice extent indicator"

_The Cryosphere, 2020_

## Referee Comment (RC2)

**An improved sea ice detection algorithm using MODIS: application as a new European sea ice extent indicator**

Joan A. Parera-Portell, Raquel Ubach, and Charles Gignac

**2 Feb 2021**

**General comments**

The authors have developed open water – sea ice classification algorithm for the MODIS imagery denoted as IceMap500. This algorithm is further development of previous IceMap250 algorithm (Gignac et al. 2017). The IceMap500 has some novel properties compared to the operational MODIS MOD29 sea ice product algorithm: conducting open water detection over areas where visibility is sufficient – even through clouds (this feature was in the IceMap250); these areas may have been masked as clouds in the MOD35 cloudmask product, and correcting deficiencies originating from the MOD35 cloudmask, e.g. from snow/ice background flag derived from the NSIDC's Near-real-time Ice and Snow Extent (NISE) product at 25 km pixel size and used in the production of the cloudmask. The accuracy of the IceMap500 algorithm was validated with manual classification of pixels randomly selected from 32 sea ice maps. The average accuracy was very good, 95.96 %, and the average kappa coefficient was 0.853.

The IceMap500 algorithm was applied to Terra and Aqua MODIS L1B/L2 swath datasets acquired in March and Sep 2000-2019 over the northern European Seas, and the resulting open water – sea ice maps are compiled to monthly sea ice extent (SIE) maps. These monthly SIE maps are used to investigate SIE trends. No significant trends were observed for the Baltic Sea, but the European Arctic seas showed clear negative trends both in March and September, which is in agreement with many previous sea ice trend studies.

I think the most interesting and important part of the study is the new IceMap500 algorithm, which seems to be very good in the open water – sea ice classification, and to perform better than the MOD29 product (although this was not explicitly discussed in the paper). It would be very interesting to see figures related to different phases of the IceMap500 algorithm shown in Figure 3. I don't quite understand why a new MODIS monthly SIE extent map is needed; there are already microwave radiometer data based products available, e.g. NSIDC sea ice index used as comparison data in the paper – what new information the MODIS monthly SIE map gives us? It delineates the sea ice edge in finer resolution than the radiometer product, but does the resolution improvement mean much in monthly scale? In this time scale there is a lot averaging of the ice edge position due to the sea ice movement. The authors should explain why the new MODIS monthly SIE map is possibly needed, and how it is better than the existing products.

For the Baltic Sea a monthly SIE map does not make much sense as it is a highly averaged map due to sea ice movement in a small sea. The Baltic Sea Ice Services and many studies uses a maximum SIE during a single day within the ice season. I think you should drop the Baltic Sea SIE study with monthly data.

You should review in more details what kind of sea ice detection algorithms and products there are for the optical imagery, what are their deficiencies, what kind of sea ice trend studies have been conducted with them, and what kind of improvements you are targeting with the IceMap500? As the previous IceMap250 algorithm is a basis for your new algorithm you could describe it in more details in Section 2.3. You could also discuss applicability of the IceMap500 to other sensors, like VIIIRS and Sentinel-3 SLSTR, possible usage of the IceMap500 maps in various applications, like ice charting, ship navigation, validation of SAR based products, and some topics for further development of the IceMap500.

In addition, you could study what is the typical minimum sea ice concentration for a pixel that it is still detected as sea ice by the IceMap500. I think this can be conducted using typical open water and sea ice signatures. Further, is bare new ice typically detected as sea ice or not? What TOA reflectances and TB values it should have to be detected as sea ice? This relates to detection of polynyas with thin ice as sea ice, and not as open water class. The effect of melt ponds would be also interesting, but I guess more difficult to quantify.

Instead of the SIE trend study, which I think do not give much information compared to previous studies, you could focus more on the capability of the IceMap500 applied on the swath data, and on daily aggregated scale, and comparison to existing product, like MOD29. Anyways, you should compare your SIE trends results (numbers) to previous studies, especially to the most recent ones, now it seems that latest studies you referred are from 2014, see e.g.

Matthews, J.L.; Peng, G.; Meier, W.N.; Brown, O. Sensitivity of Arctic Sea Ice Extent to Sea Ice Concentration Threshold Choice and Its Implication to Ice Coverage Decadal Trends and Statistical Projections. *Remote Sens.* **2020**, *12*, 807. https://doi.org/10.3390/rs12050807

Chen Ping, Zhao Jinping. 2017. Variation of sea ice extent in different regions of the Arctic Ocean. Acta Oceanologica Sinica, 36(8): 9–19, doi: 10.1007/s13131-016-0886-x

Comiso, J. C., W. N. Meier, and R. Gersten (2017), Variability and trends in the Arctic Sea ice cover: Results from different techniques, J. Geophys. Res. Oceans, 122, 6883–6900, doi:10.1002/2017JC012768.

Julienne Stroeve and Dirk Notz, "Changing state of Arctic sea ice across all seasons," 2018 Environ. Res. Lett. 13 103001

Your study area covers the northernmost European sea regions defined following the European Union's Marine Strategy Framework Directive (MSFD) with addition of 400 km buffer, as shown in Figure 1. This leads to the Kara sea being included only partially, and the northern border not following any latitude limit or natural geographical borders. I think it would be better to follow many previous sea ice studies in the definition of your study area: e.g. Greenland, Barents and Kara Seas, see NSIDC region mask (https://nsidc.org/data/polar-stereo/tools_masks.html#region_masks). This way your SIE trend results could be better compared with previous studies.

In general, I think your IceMap500 algorithm has been developed and validated properly, and it makes important addition to the tools used in the sea ice remote sensing with the optical imagery. It should be of high interest to the sea ice remote sensing community. However, there are some details missing which I point out below in specific comments. Currently, I don't clearly see value of the MODIS monthly SIE product and its usage in the SIE trend studies. I think the paper needs improvement in this issue and more detailed comparison to previous studies, or even dropping it and focusing more on the IceMap500 algorithm and usage of the swath/daily SIE product.

**Specific comments**

**Abstract**

"Aiming at the creation of an improved European sea ice extent indicator, the IceMap250 algorithm has been reworked…"

Likely a reader does not have at this point any clue what is the IceMap250 algorithm. Add at least reference if that is allowed in TC.

"…systematically achieving accuracies above 90 %."

You could mention how this accuracy figure was determined.

**1 Introduction**

line 18: "The Arctic sea ice cover has been changing rapidly over the last decades, with its overall extent declining steadily since the first satellite observations in the late 1970s (Serreze et al., 2007; Comiso et al., 2008; Cavalieri and Parkinson, 2012; Massonnet et al., 2012; Meier et al., 2014)"

You should add some newer studies as reference.

l 21: "Moreover, sea ice thickness has decreased as much as 65 % in the period extending from 1975 to 2012 (Lindsay and Schweiger, 2015)."

More recent studies here:

Kwok, "Arctic sea ice thickness, volume, and multiyear ice coverage: losses and coupled variability (1958–2018)", Environ. Res. Lett. 13 (2018) 105005

Liu et al., "Multidecadal Arctic sea ice thickness and volume derived from ice age," The Cryosphere, 14, 1325–1345, 2020.

You could also write about observed decrease of multiyear ice.

l. 32: "Several sea ice variables are continuously obtained and distributed by institutions such as the EUMETSAT Ocean and Sea Ice Satellite Application Facility (OSI-SAF) or the National Snow and Ice Data Center (NSIDC), commonly at resolutions between 10 km and 25 km."

Add references to the products.

l. 43: "as happens with MODIS sea ice products MOD29 and MYD29."

Explain what are these MODIS products; what they contain, and add references to product algorithm and to products.

**2 Materials and Methods**

You should present in this section the NSIDC's Sea Ice Index data which you use later in Section 3.3; how it is calculated, resolutions, accuracy figures, how it is processed to match your data, etc., with references.

l. 57: "This work focuses on the European regional seas established by the MSFD."

Any reference for this?

In Section 2.2 add references to MODIS instrument, datasets and dataset algorithms, e.g. to:

l. 79: "It acquires data in 36 spectral bands, ranging from the visible spectrum to the thermal infrared. Spatial resolution at nadir varies from 250 80 m (bands 1 and 2) to 500 m (bands 3-7) and 1 km (bands 8-36)."

l. 85: "consisting of MODIS Terra level 1B Top-of-Atmosphere (TOA) radiance products MOD02HKM, MOD021KM, and the MOD35_L2 cloud mask."

Spell out acronyms, like SWIR, in Table 1.

l. 89: "Although TOA data does not reflect the physical properties of sea ice and water, it avoids extensive processing due to atmospheric correction"

I don't understand this; how come the TOA reflectance does not depend on the physical properties of sea ice and water?

Describe in Section 2.2 how MODIS L1B and L2 swath datasets are georectified, i.e. gridded, to your study area shown in Figure 1. What projection is used, polar stereographic?

l. 102: "Threshold tests based on the Normalized Difference Snow and Ice Index 2 (NDSII-2)"

You could show here equation (3) for the NDSII-2.

You could mention that the Jenks natural breaks optimization is the same as k-means clustering applied to univariate data.

l. 103: "the TOA reflectance at 545-565 nm"

Give MODIS band number here.

l. 107: "When batch processing MODIS data it may be likely to run into scenes lacking either ocean water or sea ice and, consequently, the Jenks optimization splits pixels into both surface classes erroneously."

Could you prevent this by using some kind of pre-classification data to open water vs. sea ice or using co-incident MOD29 product as priori information? If these show that there is only (mostly) one surface class in the image then the Jenks optimization split is not conducted for the threshold determination, but a threshold from a similar image (same time of year and similar location) is used?

l. 128: "summarized in the product's user's guide (Strabala, 2004)."

Add also references to MOD35 product algorithm.

l. 129: "Unobstructed FOV, selecting only pixels identified as confident clear.

Add that 'confident clear' means confidence > 0.99; from Ackerman et al. (2010).

Describe in the text how the downscaling in Figure 3 is conducted.

l. 139: "This mask is intended to identify areas where visibility is sufficient to perform a classification, for the sole goal of detecting open water."

Se detection of open water is possible through thin clouds?

l. 143: "where µ and    are the mean and standard deviation of R(B20/B32),"

Are the mean and std swath calculated from the same swath data which is being classified with R?

l. 167: "Mid-range infrared has been selected instead of thermal infrared because the atmospheric correction is straightforward and may be affected by reflected solar radiation, making easier the exclusion of sun glint as a result of the temperature increase."

But you are not conducting atmospheric correction of the MODIS data, so why you mention "the atmospheric correction is straightforward"?

l. 171: "refer, for instance, to global SST products by the NOAA"

Give reference to these SST products.

Section 2.3.3 MODI35 correction: I don't quite follow how this works: you have the MOD35 map which has NISE artefacts or blocks. These blocks are marked as cloudy, so how you remove them with the MOD35 correction when the TOA data is not available; it is masked as cloudy? The correction only affects cloud-free areas close to the blocks? Please show here some example figures.

l. 216: "The synthesis maps are generated by calculating the sum of composite maps where ice = 1 and water = 0, and later normalizing the results according to the maximum number of coincident sea ice observations achieved."

Give some statistics for the maximum number of coincident sea ice observations.

l. 225: "Finally, the euclidean distance from both sea ice and water is calculated, and is later used to fill NoData gaps by setting as sea ice those pixels closer to sea ice than to water,"

What is a typical area fraction of NoData gaps in a monthly map?

**3 Results**

The Results Section could have short introduction in the beginning of its content.

l. 240-242: What p-value limit you used here?

l. 244: "While the Baltic Sea trend line in Fig. 7 clearly shows a negative tendency,"

I don't think there is a clear trend by visual analysis; seems quite 'random' variation.

l. 267: "On the contrary, as due to the extensive sea ice cover March scenes are especially prone to almost lack water"

Looking your large study area in Figure I would assume you have always open water in so,e MODIS imagery, e.g. south of Svalbard. Or do you mean that there are many scenes with only sea ice? Please elaborate.

Table 5: How about giving the percentages with 0.1% accuracy? Same for the text. Three decimals for the kappa coefficient really needed?

It seems that Figure 9 is not discussed nor introduced in the text,

**4 Discussion**

l. 324: "According to EEA (2016), Baltic sea ice extent trends are affected by large interannual variability caused by the North Atlantic Oscillation that prevents them from being statistically significant."

You could put here also Vihma and Haapala as reference. Also the Arctic Oscillation (AO) has a role here: the annual maximum ice extent generally decreases with increasing indices of AO and NAO.

The monthly March and September sea ice extent maps from 2000 to 2019 are available at https://doi.org/10.5565/ddd.uab.cat/196007.

What is the projection in the tiff-file maps? I did not find it in the readme file.

---

## Author Comment (AC1)

**Response to reviewer #1**

**"An improved sea ice detection algorithm using MODIS: application as a new European sea ice extent indicator", by Joan A. Parera-Portell, Raquel Ubach, and Charles Gignac**

Dear reviewer #1,

First of all thank you for your comments and suggestions, which will surely help improve the current manuscript. Here it is a detailed answer for each of your **major comments**:

1) I think you are right, there are many references to Gignac et al. (2017) that could be avoided. I also agree with the excessive use of "we" in the introduction. However, I would like to clarify that we are not the IceMap250 team: we independently tested IceMap250 and developed IceMap500 with Dr. Gignac as a collaborator. Thus, answering to one of your specific comments, we dropped the 250 m downscaling for various reasons: I) simplicity, II) reduced processing times, III) problems with MODIS Aqua band to band registration reported by various authors, and IV) spectral integrity of the imagery (since no downscaling is applied).

2) In my opinion, the most important feature of IceMap500 is the way it diminishes the effect of the NISE footprint, which really hinders the mapping of the sea ice edge (see Fig.2 in the manuscript). Therefore its major benefit is the increase of mapped area respective to MODIS MOD29/MYD29 sea ice extent products and IceMap250 itself, which nonetheless has a higher resolution. The maps generated by IceMap500 covering the coastline or the ice edge are very detailed and consistent in this sense and, as you say, it should be further explained and exemplified. In this regard, in Figure A are shown two different monthly composites centered in different regions where the resolution of other sea ice extent datasets might be too coarse.

The maps in Figure A are also useful to explain the derivation of our monthly extent maps from monthly sea ice presence likelihood maps, as readers might wonder why the two do not perfectly match when overlaid. As said in the Methodology section, pixels where sea ice presence is $>0\,\%$ and $<10\,\%$ are discarded because such observations might not be reliable enough ($0\,\%$ is water). In practice, what happens is that usually by eliminating such observations one gets a small NoData buffer zone along the ice edge. We then take advantage of the pixels set as water (remember that the sole goal of the VIS mask is to detect open water) and fill the gaps using an Euclidean distance allocation algorithm. This way we get a clearer and smoother sea ice edge, which nonetheless does not completely ignore the information carried by pixels where likelihood $<10\,\%$.

3) Yes, September 2013 is an interesting case and it can further demonstrate the accuracy of IceMap500 when detecting fragmented ice. Figure B compares the Sea Ice Index (SII) and IceMap500 monthly maps corresponding to Sept. 2013, which I think would be a useful addition to the manuscript. What we found is that IceMap500 classifies way more ice due to the presence of fragmented ice and ice floes, which are not seen in the SSI due to the monthly mean sea ice concentration not exceeding $15\,\%$ within those cells (remember that the cell size is 25 km). Therefore, differences do not only come from the coarser resolution of the SSI, but also from the different ways in which the SSI and IceMap500 obtain the monthly extent: while the SSI calculates extent from the monthly mean concentration,

[Figure]

Figure A: IceMap500 monthly extent maps covering different regions of the Arctic and Baltic seas. In the right panels the monthly sea ice presence likelihood is shown.

the IceMap500 monthly maps may be better understood as the maximum sea ice extent achieved during that month (considering the 10 % sea ice presence likelihood threshold).

4) Indeed, processing time is a major concern regarding any computer algorithm and the manuscript would benefit from including details on this matter. The time required to generate a sea ice extent map for a given scene depends on many factors, including how many pixels there are inside the study area, the amount of clouds and nighttime areas, the number of pixel samples used to calculate the Jenks breaks and the time that NASA's HDF-EOS to GeoTIFF Conversion Tool takes to project the original hdf files. I have run IceMap500 on multiple computers, from laptops to desktop PCs but, to illustrate the processing time, consider Figure C. This is a fairly large scene with both surface classes, sea ice and water. Currently, on a desktop computer under Linux Mint 20.1, 2.67GHz $\times$ 4 processor and 12 Gb RAM IceMap500 takes about 230 s to create the map seen in Fig. C (excluding the data download time). A full daily map covering our study area (16 scenes) took 50 min to process. Of course, the processing time would become smaller with more powerful machine specifications .

[Figure]

Figure B: Comparison between IceMap500 monthly map and the Sea Ice Index (September 2013). In the right panel the monthly sea ice presence likelihood is shown.

Finally, I would like to thank you for your text corrections and to answer some of your **specific comments**:

l.3: The reference to IceMap250 will be dropped, as it has been also suggested by reviewer #2.

l.14: Indeed this sentence is unclear and will be rewritten. What it really means is that the areas that we compare are not accounting for the error in the trend lines.

l.32: We will, as you suggest, provide more background about sea ice remote sensing.

l.40-45: see my general comment n.1. We will fix the text so it becomes more clear to the readers.

l.231: You are right, an introductory text should improve the transition between sections. We will probably also add subheadings so the structure of the discussion becomes more clear.

l.300: The agreement between the NSIDC's SSI and IceMap500 depends on the number of pixels that are classified as sea ice in both datasets, i.e. where both datasets agree there is sea ice. Plainly put, if there is an agreement of 90 % this means that the SSI and IceMap500 overlap on the 90 % of sea ice pixels. The other 10 % are pixels that are classified as sea ice only by either one of the maps. The difference in spatial resolution is not compensated, as the sole goal of this agreement analysis is to show the consistency of IceMap500. What is seen in Figure 8 is that agreement in March is very stable, only slightly oscillating around 90 %. This suggests that besides the difference in resolution, there are no significant differences between IceMap500 and the SSI, even considering the different monthly extent calculation approach. Instead, in September the mean agreement decreases while its variability greatly increases, so there is something more going on: sea

ice is considerably more fragmented, which makes the effects of the resolution and the extent calculation to be much more important (see Figure B).

Figure 9: We plan to include a graphic comparison with other sea ice products. The figure will surely include a more zoomed in comparison with the SII, as you suggest.

[Figure]

Figure C: Single classified swath example (Arctic).

---

## Author Comment (AC2)

**Response to reviewer #2**

**"An improved sea ice detection algorithm using MODIS: application as a new European sea ice extent indicator", by Joan A. Parera-Portell, Raquel Ubach, and Charles Gignac**

Dear reviewer #2,

Thank you for your constructive comments, they will certainly help us in improving the quality of our manuscript. In the first place I will answer your **general comments**:

Overall, I agree that the most important part of the study is the algorithm itself, not the monthly trends. The trends (and their agreement with previous studies that you point out) are just a demonstration of the algorithm's usefulness and reliability. However, we want to stress that the IceMap500 raw products are the swath map and the daily map, so our monthly maps are, essentially, a way of capturing the trends in the study area. Thus, there is averaging of the sea ice edge indeed, and here is where both the sea ice presence likelihood maps and the comparison to the Sea Ice Index (SSI) gain importance.

In the first case, the sea ice presence likelihood maps add valuable information to the monthly sea ice extent (SIE), such as how many times sea ice was detected in a certain area or pixel. This allows us to detect the places where sea ice (and also the ice edge) has been more unstable during that month, as the sea ice presence likelihood will drop (see Figure A). In fact, the likelihood maps allow even to detect cracks in the sea ice, and of course if sea ice has moved significantly the sea ice presence likelihood will be lower. Under a certain likelihood threshold, which is 10 % in our case, we set the pixels as NoData because they may not be reliable enough, and this usually leaves a small NoData buffer zone along the ice edge (see Figure B) where sea ice presence likelihood is >0 % and <10 % (0 % is water). It is within this buffer zone that we place the sea ice edge during the monthly SIE map creation, as we use an Euclidean allocation method to set those pixels either as water or sea ice. This generates a smoother sea ice edge, and should be considered as the maximum extent achieved during a given month.

On the other hand, the SII-IceMap500 monthly SIE comparison allows to assess where do our SIE maps agree or disagree with a commonly used index. The agreement analysis indicated that one of the main differences is the capability of IceMap500 of detecting small sea ice floes and fragmented ice and incorporating them into the monthly SIE map, especially in September. In March the SII and IceMap500 datasets are very similar, so even with the much greater spatial resolution of IceMap500 and the different SIE derivation approach they agree very well. Of course, IceMap500 was never intended to replace SII or other datasets, but instead to provide additional and higher resolution information which could be interesting for local or regional studies of sea ice conditions (the European sea regions, in our case). In our opinion, the agreement with the SII indicates that our maps, even the monthly SIE maps, are coherent and reliable. Some strong features of our monthly SIE maps are the increased classified area in comparison to MOD29 (which increases the sea ice presence likelihood values), and the more detailed SIE information along the shoreline, fjords and other areas that sensors with a coarser resolution might miss. Thus, we think that IceMap500 represents an improvement towards local and regional sea ice studies even at a monthly scale, especially taking into account the spatiotemporal information that the sea ice presence likelihood maps may provide, and is complementary

[Figure]

Figure A: IceMap500 monthly sea ice presence likelihood map of the Baltic sea.

to informations commonly gathered by national sea ice services for their operational ice conditions monitoring activities. Also, the likelihood maps can be generated within any given time period, so it might be a useful approach to analyze, for instance, weekly sea ice presence, and a more conservative threshold can be used to obtain SIE. Additionally, the moderate processing time of the algorithm should allow to produce datasets that could be of interest for sea ice services, marine infrastructure managers or for navigation, as a full daily map covering our study area (16 scenes) takes about 50 min to process in a machine under Linux Mint 20.1, with a 2.67GHz $\times$ 4 processor and 12 Gb RAM.

We think that, as you suggest, a revised manuscript would greatly benefit from a subsection focused on comparing daily maps from different sources, such as MOD29 and EUMETSAT Daily Sea ice Edge. Therefore, the effect of the NISE footprint could also be further discussed. Illustrating the different phases of the algorithm would be also an improvement in the Methodology section, as you say, and it would surely help readers understand how it works step by step. We plan to add this to the manuscript in the revision. We also plan to compare the SIE trends in the papers you have suggested, which we unfortunately missed. However, as the original goal of the algorithm was to provide complementary and higher resolution information for existing sea ice extent indicators used in the European Union, we plan to keep our current study area. In addition, the trends in our study are intended to proof the algorithm's fitness by comparing them to existing ones, but it is not intended to be an exhaustive trend analysis, as our goal is to demonstrate that IceMap500 can be used to monitor sea ice at a European scale level.

The applicability of IceMap500 to other sensors is also another topic which deserves further discussion and which would greatly enhance the manuscript. Most MODIS bands we use in IceMap500 have their equivalent in both VIIRS and Sentinel-3 SLSTR at almost the same wavelength ranges, with the only exception of MODIS band 7 (2.105-2.155 µm) which we use as a cloud detector in the MOD35 correction. The closest matches in VIIRS

[Figure]

Figure B: Detail of IceMap500 monthly sea ice presence likelihood map, showing areas where likelihood is $>0\,\%$ and $<10\,\%$.

and Sentinel-3 SLSTR have central wavelengths of approximately $2.250\,\mu m$, a region in which ice has a reflectance peak of about $20\,\%$, while at $2.105\text{-}2.155\,\mu m$ it has $5\,\%\text{-}10\,\%$ reflectance. It would be worth investigating 1) whether VIIRS and Sentinel-3 SLSTR require such artifact correction and 2) if they do, whether clouds and sea ice can still be distinguishable by their reflectance at $2.250\,\mu m$.

Finally, as for bare ice and thin ice, the thresholds in IceMap500 are designed to detect those surfaces as sea ice too. The $17\,\%$ ToA threshold using band 4 (545-565 nm) had been previously used in Riggs et al. (1999) and Gignac et al. (2017) and is intended to include most low-albedo sea ice. Validation in Gignac et al. (2017) clearly shows that the B4$>=17\,\%$ threshold resides slightly into the upper standard deviation of the water class reflectance, so the risk of misclassifying melt ponds, new ice, leads and polynyas is low (see, for instance, Figure 6 in the Gignac et al. (2017) paper). The Normalised Snow and Ice Index 2 (NDSII-2) test also is shown to discriminate 96-100 $\%$ of the sea ice even during the melting periods in Gignac et al. (2017). The band 20 temperature threshold ($1\,°C$) is intended to be a mask and not really a classification test, so it generates a buffer zone that performs pretty well including both cold water and new sea ice. Past studies such Zhang et al. (2017), already cited in our manuscript, show that melt ponds stay below $0.3\,°C$, so the threshold should be safe. However, both the NDSII-2 test and the band 4 reflectance test may still fail, as our validation results demonstrate. A strength of the MOD35 correction is that, if one of the threshold tests fail, those pixels are set as NoData and may be classified again during the correction. Then those pixels have a higher probability of being classified as sea ice, as the area in which the NDSII-2 test is applied is smaller and so the Jenks threshold tends to include more low-albedo ice. That is, always when ToA reflectance at band 4 is $17\,\%$ or above.

**Specific comments**

**1) Abstract**: I think we should drop the reference to IceMap250 and just explain what it is later in the manuscript. Also, we will mention the validation method we used to test the accuracy.

**2) Introduction**: We will take a look at the references you suggest and include them to update the sea ice information. References to the mentioned sea ice products will also be included, also with a brief description of each product.

**3) Materials and Methods**: all the references and text corrections you suggest will be added to the manuscript, including the NDSII-2 equation. Now, regarding your more technical questions:

l.89: This line is not clear enough and will be rewritten. It does not really mean that TOA reflectance does not depend on the physical properties of sea ice and water, but instead that TOA reflectance is not itself a physical property of sea ice and water because of the atmospheric contribution, although it is obviously related to the surface reflectance.

Section 2.2: We will explain step by step the workflow of the algorithm, including what you suggest here. For projection purposes we use NASA's HDF-EOS to GeoTIFF Conversion Tool. Also, we unfortunately did not mention the projection we use both in our figures and the algorithm itself, which is North Pole Lambert Azimuthal Equal Area.

l.107: What you suggest is an interesting approach, although we did not consider a solution like that because we wanted to use as few input data as possible for the sake of simplicity and efficiency. What we did is increasing the restrictiveness of the classification: even though a pixel passes the NDSII-2 Jenks optimization test, it still has to be confirmed as sea ice by the 17 % band 4 ToA reflectance threshold. Then, if water erroneously passes the NDSII-2 Jenks optimization test, it should be automatically discarded due to its low reflectance. This increase in restrictiveness does not suppose a significant change in the final classified sea ice area, because as we discussed earlier both tests are designed to include low-albedo sea ice.

l.139: Yes, it is intended to do so. This way the mapped area is considerably larger.

l.143: You are right, the mean and std are calculated every time from the swath data that is being processed.

l.167: We are not conducting atmospheric correction on the solar reflective data, nor on the thermal bands when calculating the VIS mask. However, band 20 is indeed atmospherically corrected, but only when performing the SST test. This way it is much easier to select a temperature threshold.

Section 2.3.3: Even though the area occupied by the NISE artefacts in the MOD35 mask is tagged as cloudy, the algorithm still has access to unmasked TOA reflectance so these areas can be analysed again. What IceMap500 does is to create a 25 km buffer zone around the areas classified as sea ice in the MOD35 mask. It then masks again the original TOA reflectance data using only the buffer zone, while a new cloud mask within this buffer is created using band 7. So, actually, the MOD35_L2 cloud mask is ignored within the areas that the MOD35 correction classifies. We plan to add a step-by-step graphic example in the revision, so I hope this process will become more clear to readers.

l.216: Unfortunately I cannot give you exact numbers, as the algorithm directly calculates sea ice presence likelihood in % without saving such information. However, we

can estimate the maximum number of observations by assuming that the minimum is 1 (excluding water, that is, 0) and using the corresponding % to extrapolate the maximum. We get that the mean maximum number of sea ice observations is 50 in March and 62 in September.

l.255: NoData gaps tend to appear in the northernmost regions of our study area in the March monthly maps. This is a consequence of the poor lighting conditions during the winter months; remember that we only keep pixels tagged as day in the day/night flag of MOD35_L2. Obviously, this also makes the sea ice presence likelihood to drop. September has no such lighting limitations, so NoData gaps appear more randomly and are fundamentally linked to the cloud cover (see the examples in Figure C). Overall, the mean NoData area fraction of our monthly time series is 1.0 % in March and 0.7 % in September. However, March features a larger std (0.7 %) compared to the std of September (0.3 %).

[Figure]

Figure C: IceMap500 monthly sea ice presence likelihood map of March 2019 (left) and September 2019 (right).

**4) Results**

l.240-242: With typical p-values of 0.05 and 0.01 our Arctic trend lines remain statistically significant. This information was accidentally omitted in the text.

l.244: You are right, both the visual and the quantitative analysis do not show any clear trend in the Baltic, so this line will be removed.

l.267: Indeed most scenes feature both surface classes. However, in March the extensive sea ice cover plus the presence of clouds may cause the classified water area to drop considerably in some scenes, especially to the east of Novaya Zemlya and Franz Josef Land. Even though there may still be some pixels classified as water, the area fraction compared to sea ice is very small and so random points used for accuracy assessment may not sample those water areas, causing kappa coefficients to drop.

Table 5: Right, we can give percentages with $0.1\,\%$ accuracy and reduce the decimals of the kappa coefficients from three to two.

Figure 9: Also right, we accidentally did not reference the figure in the text.

**5) Discussion**: We will include the reference you suggest and discuss on the effect of the Arctic Oscillation. As for the projection of the maps in the dataset (`https://doi.org/10.5565/ddd.uab.cat/196007`), you are right, there is no information about it in the readme file. We will amend this issue as soon as possible. All maps are in North Pole Lambert Azimuthal Equal Area.

---

## Author Response (AR1)

**Author response**

**"An improved sea ice detection algorithm using MODIS: application as a new European sea ice extent indicator", by Joan A. Parera-Portell, Raquel Ubach, and Charles Gignac**

Dear editor and referees,

On behalf of all co-authors I want to thank you for the helpful advice and comments you provided. I hope you find the revised manuscript a substantial improvement from the initial submission. This document is divided in two sections: first the editor's comments are answered, and then the major changes in the revised manuscript are summarized.

**Response to editor's comments**

-The caption of Figure 1 and line 102 have been corrected.

-Regarding the VIS mask threshold value, clear areas are those where VIS < 0.5. I assume this question is because in the description of the previous IceMap250 (Gignac *et al.*, 2017) it is written as VIS > 0.5, which Dr. Gignac confirms is a typo.

-Table 5 (now Table 4) has been changed so monthly commission and omission errors are shown to support the results in the text. The caption has also been rewritten, so now it indicates what kappa coefficients are and why there are two accuracy results per month. Each number corresponds to a different scene, so four scenes were validated each year (two per month).

-Much of the manuscript has been rearranged. Now inconsistencies between IceMap500 and the Sea Ice Index are discussed in a more detailed way, and a September 2013 figure is included. We also discuss the effect of the sea ice presence likelihood threshold on the agreement between both datasets.

**Changes in the manuscript**

-Abstract: minor corrections and changes. IceMap250 is not cited before introducing it. The last sentences, which the referees found confusing, were also changed.

-Introduction: the referees found this section poorly structured and lacking some recent references, so it has been largely rewritten and updated. A brief summary of sensors used in sea ice remote sensing and an introduction to MODIS is included. Limitations of existing MODIS sea ice extent products are also highlighted.

-Materials and Methods: minor changes in subsection 2.1 (Study area), related to the subdivision of the study area in Baltic and NE Atlantic-Barents regions for the sea ice trends analysis. Subsection 2.2 (selected data) has been reduced, as most text has been moved to the Introduction. A new section 2.3 (Overview of previous MODIS sea ice extent algorithms) is included where IceMap and especially IceMap250 are introduced, thus making easier to explain the workflow of IceMap500 in later sections. Subsection 2.4

(IceMap500: challenges and improvements) was rearranged due to the new subsection 2.3. Minor changes and corrections were done in the text where the masking, the classification tests and the MOD35 correction are explained. However, the text explaining the map aggregation method has been expanded, including two new figures: a step-by-step example of IceMap500 and a four-panel figure with sea ice presence likelihood examples already shown in the responses to the reviewers.

-Results: the former Figure 7 and Table 4 (sea ice trends) have been merged in a single figure. Table 5 (now Table 4) has been expanded as indicated in the response to the editor's comments. An equation for IceMap500-Sea Ice Index agreement is now given as suggested by the referees.

-Discussion: this section has been rearranged and considerably expanded. Now it consists of two subsections (4.1 Sea ice trends and 4.2 Applicability of IceMap500). In 4.1 a more detailed comparison of the sea ice trends is given, including more recent references and numerical comparisons. In 4.2 we discuss the major error sources of the algorithm and the effect on the overall accuracy and on the IceMap500-Sea Ice Index agreement. We also discuss the potential of IceMap500 as a sea ice monitoring tool, taking into account its resolution and processing times. We also compare the results of IceMap500 to similar existing products (including a figure), and discuss the application of IceMap500 to other sensors.

-Conclusions: minor corrections.